# New Fossil Xyelidae (Hymenoptera: Symphyta) from the Mesozoic of Northeastern China [note 1]

**DOI:** 10.3390/insects13040383

**Published:** 2022-04-13

**Authors:** Liyang Dai, Alexandr P. Rasnitsyn, Chungkun Shih, Mei Wang, Dong Ren

**Affiliations:** 1College of Life Sciences and Academy for Multidisciplinary Studies, Capital Normal University, 105 Xisanhuanbeilu, Haidian District, Beijing 100048, China; dailiyang291@163.com (L.D.); chungkun.shih@gmail.com (C.S.); 2Key Laboratory of Forest Protection of National Forestry and Grassland Administration, Institute of Forest Ecology, Environment and Nature Conservation, Chinese Academy of Forestry, Beijing 100091, China; 3Paleontological Institute, Russian Academy of Sciences, 123 Profsoyuznaya ul., 117647 Moscow, Russia; alex.rasnitsyn@gmail.com; 4Department of Palaeontology, Natural History Museum, Cromwell Road, London SW7 5BD, UK; 5Department of Paleobiology, National Museum of Natural History, Smithsonian Institution, Washington, DC 20013-7012, USA

**Keywords:** Xyelidae, Macroxyelinae, fossil insects, new species, China

## Abstract

**Simple Summary:**

One new genus and two new species, described from new fossil specimens of northeastern China, enhance our knowledge of the Xyelidae in the mid-Mesozoic. A key to the genera of Angaridyelini and a table of known fossil species of Macroxyelinae are provided. After investigating various angles between Rs+M and 1-Rs for known fossil species of Macroxyelinae, we report that the angle and the length of 1-Rs are correlated. In addition, we believe that the tribe Ceroxyelini may have only one genus of *Ceroxyela*, and suggest that *Isoxyela* and *Sinoxyela* should be transferred to Gigantoxyelini.

**Abstract:**

One new genus and species, *Leptoxyela eximia* gen. et sp. nov., and one new species, *Scleroxyela cephalota* sp. nov., are described and illustrated based on two well-preserved compression fossils from the Lower Cretaceous Yixian Formation and the Middle Jurassic Jiulongshan Formation of China, respectively. *Leptoxyela eximia* gen. et sp. nov. is placed in the tribe Angaridyelini, 1966 and *Scleroxyela cephalota* sp. nov. in the tribe Xyeleciini Benson, 1945; while both tribes are in the subfamily Macroxyelinae Ashmead, 1898 of Xyelidae. A key to the genera of Angaridyelini is provided. In addition, we investigated various angles between Rs+M and 1-Rs for known fossil species of Macroxyelinae, and we found the angle and the length of 1-Rs are correlated; however, we could not see any correlation between the angles and the fossil ages even within a tribe. Furthermore, based on Sc_2_ connected to R before Rs, the angle between Rs+M and 1-Rs, and the length of the first flagellomere, we believe that the tribe Ceroxyelini may have only one genus of *Ceroxyela*, and suggest that *Isoxyela* and *Sinoxyela* should be transferred to Gigantoxyelini.

## 1. Introduction

Hymenoptera, one of four mega-diverse insect orders, comprises more than 155,000 described species [1,2] and plays a fundamental role in almost all terrestrial ecosystems [3,4]. Hymenoptera consists of the suborders of Apocrita and Symphyta [5]. Symphyta include about 9000 species in the world [6]. The most important diagnosis trait of Symphyta is the presence of broad waists, contrary to the Apocrita. As the most basal group of Symphyta, the small family Xyelidae has a special status in the evolutionary history of the Symphyta and Hymenoptera [7,8,9].

The earliest appearances of xyelids have been reported from the Middle or Upper Triassic of Kyrgyzstan [10], and the Upper Triassic of Australia [11,12], South Africa [13], Argentina [14] and Japan [15]. The family reached its maximal of diversity between the Middle Jurassic and the Early Cretaceous, and its past distribution was much broader than today [16]. There are four subfamilies in the Xyelidae: Archexyelinae Rasnitsyn, 1964; Macroxyelinae Ashmead, 1898; Madygellinae Rasnitsyn, 1969 and Xyelinae Newman, 1834 [17], with numerous fossil records. Hitherto, more than 85 species within 50 genera belonging to four subfamilies of Xyelidae have been reported in the Mesozoic and Cenozoic [18,19,20,21].

Macroxyelinae are a common subfamily of Xyelidae in the Mesozoic and can easily be differentiated from all other subfamilies because of the following combination of characters: the sclerotized apical bridge between C and R_1_ next to pterostigma, short and wide ovipositor and at least partially sclerotized pterostigma [10]. Macroxyelinae encompass five tribes: Angaridyelini Rasnitsyn, 1967; Ceroxyelini Rasnitsyn, 1969; Gigantoxyelini Rasnitsyn, 1969; Macroxyelini Ashmead, 1898 and Xyeleciini Benson, 1954 [22]. Up to date, 28 genera containing 44 fossil species have been reported in Macroxyelinae (Table 1).

In this report, we describe a new genus and species as well as a new species of a previously erected genus *Scleroxyela*, which enhance our knowledge of the Xyelidae in the Mesozoic of northeastern China.

## 2. Materials and Methods

The holotype specimen of *Leptoxyela eximia* gen. et sp. nov. (Figure 1) was collected from the Lower Cretaceous, Yixian Formation; Huangbanjigou, Chaomidian Village, Shangyuan Township, Beipiao City, Liaoning Province, China. The holotype specimen of *Scleroxyela cephalota* sp. nov. (Figure 2) was collected from the latest Middle Jurassic, Jiulongshan Formation; Daohugou Village, Shantou Township, Ningcheng City, Inner Mongolia, China. All specimens are housed at the fossil collection of the Key Lab of Insect Evolution and Environmental Changes, at the College of Life Sciences and Academy for Multidisciplinary Studies, Capital Normal University (CNUB; Dong Ren, curator), in Beijing, China.

The photographs were taken with a Nikon SMZ 25 and an attached Nikon DS-Ri 2 digital camera system. Line drawings were prepared using Adobe Illustrator CC and Adobe Photoshop CC. Statistical analyses were performed using R v.4.0.3 (R Core Team, Auckland, New Zealand) The wing venation nomenclature used in this paper follows Huber and Sharkey (1993) [23]. Antennal thread means flagellar segments behind the enlarged first flagellar segment. For wing venation, 1-Rs and 2-Rs refers to the 1st and 2nd segments of Rs; 1-M means the 1st segment of M and 1-Cu, 2-Cu and 3-Cu mean 1st, 2nd and 3rd segments of Cu.

## 3. Results

Systematic palaeontology.

Order: Hymenoptera, Linnaeus, 1758.

Suborder: Symphyta, Gerstaecker, 1867.

Superfamily: Xyeloidea, Newman, 1834

Family: Xyelidae, Newman, 1834

Subfamily: Macroxyelinae, Ashmead, 1898

Tribe: Angaridyelini, Rasnitsyn, 1966

Genus: *Leptoxyela*, Dai, Rasnitsyn, and Wang gen. nov.

Type species: *Leptoxyela eximia*, Dai, Rasnitsyn, and Wang sp. nov.

Etymology: the generic name is a combination of Greek “*Lepto-*”, meaning thin and referring to the long and thin terminal part of flagellum, and the generic name *Xyela*.

Diagnosis: Antenna with thread, longer than first flagellomere. Pterostigma sclerotized basally; 1-Rs slightly longer than half 1-M, shorter than half 2-Rs; Sc_1_ connected to C beyond level of Rs and Sc_2_ connected to R just before origin of Rs; Rs_1_ closer to pterostigma than to 1-Rs_2_ along wing margin; 1m-cu apparently longer than 2-Cu; ovipositor sheaths short and moderately wide.

Species included: type only.

Remarks: *Leptoxyela* gen. nov. is attributed to Macroxyelinae, based on the sclerotisation connecting C and R before pterostigma; short and moderately wide ovipositor sheaths, and further to Angaridyelini based on pterostigma sclerotized only basally. Within Angaridyelini, the new genus is similar to *Nigrimonticola* in having long antennal thread; Sc_1_ connected to C beyond level of Rs and Sc_2_ connected to R just before origin of Rs. It can be differentiated from *Nigrimonticola* by its longer 1m-cu (see Key below for details).


**Key to the genera of Angaridyelini**
1. 1-Rs longer than 2-Rs ..................................................................................................................*Ophthalmoxyela* Rasnitsyn, 1966.
1-Rs shorter than 2-Rs (Figure 3A) ............................................................................................22. Cell 1mcu twice as long as pterostigma. Sc lost ......................................................................*Baissoxyela* Rasnitsyn, 1969.
Cell 1mcu at most 1.5 × as long as pterostigma (Figure 3F). Sc distinct ...............................33. Sc_2_ connected to R just at Rs base .............................................................................................4
Sc_2_ connected to R before Rs base (Figure 3A,C,F) .................................................................54. Pterostigma sclerotised only basally (Figure 3C) ...................................................................*Lethoxyela* Zhang and Zhang, 2000.
Pterostigma sclerotised basally and far along fore margin ...................................................*Liaoxyela* Zhang and Zhang, 2000.5. Sc_1_ connected to C before level of Rs (Figure 3A). Gena with distinct lateral horn ...........*Ceratoxyela* Zhang and Zhang, 2000.
Sc_1_ connected to C at or beyond level of Rs. Gena with no lateral horn ...................................66. Antennal thread shorter than the length of segments 1–3 combined ......................................*Angaridyela* Rasnitsyn, 1966.
Antennal thread longer than the length of segments 1–3 combined (Figure 4D,E) ..........77. 1m-cu half as long as 3-Cu ......................................................................................................*Nigrimonticola* Rasnitsyn, 1966.
1m-cu almost as long as 3-Cu (Figure 1D) ............................................................................*Leptoxyela* gen. nov.

*Leptoxyela eximia* Dai, Rasnitsyn, and Wang gen. et sp. nov.

Material: CNU-HYM-LB-2021002, housed at the College of Life Sciences and Academy for Multidisciplinary Studies, Capital Normal University.

Etymology: the specific name is from Latin “*eximius*”, meaning exceptional and referring to the well-preserved fossil specimen.

Locality and horizon: Lower Cretaceous, Yixian Formation; Huangbanjigou, Chaomidian Village, Shangyuan Township, Beipiao City, Liaoning Province, China.

Diagnosis: as for the genus.

Description (Figure 1): Female sawfly in latero-ventral view with incomplete head and wings. Body length 10 mm (excluding antennae), forewing (as preserved) 8 mm long and antenna 5.7 mm long. Head and thorax brown, legs pale, antenna and abdomen of coloration intermediate between that of the head and that of the legs, legs and antenna with some darker spots.

Head distorted. Antenna very long, scape ca. 5× as long as wide, almost half as long as first flagellomere, pedicel somewhat longer than wide, first flagellomere very thin, as long as cell 1mcu, antennal thread about 1.7× as long as first flagellomere, 1.1× as long as segments 1–3 combined, with 20 segments visible, each three times as long as wide.

Thorax distorted, with no important characters available.

Forewing with pterostigma sclerotized basally; costal area broad, Sc closer to R than to C, sub-parallel to R, with two branches, Sc_1_ connected to C beyond level of Rs and Sc_2_ connected to R just before origin of Rs; R only slightly curved; 1-Rs slightly longer than half the length of 1-M, at most half the length of 2-Rs; 1r-rs slightly longer than 2r-rs, inclined toward wing base, 3-Rs arched posteriorly; Rs_1_ closer to Rs_2_ than to pterostigma along wing margin; 1m-cu almost as long as 3-Cu and about 1.2× as long as 2-Cu; 1-Cu curved in middle. Cell 1rm long and narrow, significantly longer than 1r; cell 1r about 1.6 times as long as wide and longer than cell 2r; cell 2r trapezoidal; length ratio of cells 1r:2r = 5:3; cell 1mcu nearly as long as cell 1r; cell 2mcu longer and wider than 1mcu, about 2.2× as long as wide and 1.1× as long as cell 2rm; 2rm as long as cell 3rm; cell 1cua and cell 1a incomplete. Hind wing only fragmentarily preserved.

Foreleg and middle leg partly preserved, but structure of each part of hind leg well-preserved; foreleg with coxa thin as preserved; mesopleuron and metapleuron partly preserved; hind leg with coxa elongate, about 2.1× as long as wide; trochanter small and trapezoidal, trochantellus clearly visible; femur thick and slightly wider medially, about 3.3× as long as wide; hind tibia long, about 1.4× as long as femur, and with long and sharp terminal spur; tarsus incomplete (many segments of tarsi preserved).

Abdomen with nine segments visible, ovipositor sheath broad and short; first valvifers roughly triangular; second valvifers large, longer than first valvifers; ovipositor well preserved, first valvula as wide as second valvula.

Tribe: Xyeleciini, Benson, 1945

Genus: *Scleroxyela*, Zheng, Hu, D. Chen, J. Chen, Zhang, and Rasnitsyn, 2021

Type species: *Scleroxyela daohugouensis*, Zheng, Hu, D. Chen, J. Chen, Zhang, and Rasnitsyn, 2021.

Species included: type species and *Scleroxyela*
*cephalota*, Dai, Rasnitsyn, Shih, and Wang sp. nov.

*Scleroxyela cephalota*, Dai, Rasnitsyn, Shih, and Wang sp. nov. 

Material: CNU-HYM-NN-2021004, housed at the College of Life Sciences and Academy for Multidisciplinary Studies, Capital Normal University.

Etymology: the specific name is from Greek “*kephalon*”, meaning head and referring to the large head.

Locality and horizon@ latest Middle Jurassic (late Callovian), Jiulongshan Formation; Daohugou Village, Shantou Township, Ningcheng County, Inner Mongolia, China.

Description (Figure 2): Male (body length 5.47 mm) sawfly in ventral view with nearly complete forewings. Body and antenna dark, legs pale with some darker spots. Head (width 1.45 mm, length 0.9 mm) massive, eye large, about a quarter of head width. Antenna incomplete, with first flagellomere slightly longer than thread.

Thorax and head nearly equal in width. Medial mesoscutal line short, not extending behind notauli; notauli V-like; mesoscutellum wider than prescutum, about 1.4× as wide as mesoscutum; cenchri slightly shorter than wide; metascutum triangular-shaped; metapostnotum slender and remaining part of the thorax not discernable.

Forewing (6.42 mm long) with pterostigma completely sclerotised, costal area obviously widened proximad base of Rs; Sc with two branches, posterior branch of Sc (Sc_2_) short, subvertical to R, shorter than 1-Rs and connected to R well before origin of Rs; anterior branch (Sc_1_) long, reaching C beyond origin of Rs; R distinctly bent before origin of Rs and greatly thickened before pterostigma; 1-Rs half the length of 1-M; 2r-rs ca. 1.5× as long as 1r-rs; 3-Rs arched posteriorly; 1m-cu half the length of 3-CuA, roughly equal to 1cu-a; 3r-m about 1.3× as long as 2r-m; M+Cu slightly curved in its basal third; CuP straight, extended to the top of cell 2a. Cell 1r about twice as long as 2r and 2.6× as long as wide; cell 2r trapezoidal, length ratio of cells 1r:2r = 2:1; cell 1cua narrow and long, distinctly broader at M base, about 7.5 times as long as wide, and twice as long as cell 1mcu; cell 2mcu long and narrow, longer than 2rm and about 3.6 times as long as wide; cell 1a 1.75 times as long as 2a.

Hind wing with 1-Rs vanishing short (1r-m meeting Rs close to Rs base); limit between 1-M and 1r-m obscure; M+Cu nearly straight.

Foreleg and middle leg incomplete, hind leg with coxa elongate, about 1.5× as long as wide; trochanter small and trapezoidal; femur oval, slightly wider medially, about 4.6× as long as wide; hind tibia long, about 1.1× as long as femur; tarsi not preserved.

Abdomen with nine segments visible, genitalia visible, hypopygium large, diamond-shaped; gonocoxa wide, subtriangular (narrowed basally), apparently wider than gonostylus; gonostylus long, with distinct gonomacula.

Remarks. *Scleroxyela cephalota* sp. nov. can be assigned to the subfamily Macroxyelinae based on the pterostigma slerotised and with sclerotisation connecting C and R near pterostigma, to the tribe Xyeleciini based on Sc_2_ connected to R before Rs and 1-Rs at most half the length of 1-M, and to the genus *Scleroxyela* based on pterostigma completely sclerotized and Sc_1_ reaching C distinctly behind Rs. Within the genus, the new species differs from *S. daohugouensis* in having the head wider, pterostigma wider than 1r-rs length, Sc_2_ connected to R more distant from Rs base, and forewing shorter (6.42 mm long vs. 11.1 mm in *S. daohugouensis*). These differences are unlikely to be sexually dependent.

## 4. Discussion

We describe one new genus and two new fossil species, from Huangbanjigou and Daohugou of China, respectively. Both Daohugou and Huangbanjigou are important insect deposits containing abundant, well-preserved insect fossils [24]. The Daohugou area, an important locality for the Middle Jurassic insects, located in northeast of China, is assumed to be located in a humid and warm–temperate climate area by the paleoenvironment reconstruction analysis. The Huangbanjigou area, as an important section of the Yixian Formation, also has a large number of rare animal, insect and plant fossils [25]. However, different experts have different views on the age of the Yixian Formation. Three opinions have been offered: Late Jurassic [26], transition from Late Jurassic to Early Cretaceous [27,28], and Early Cretaceous [29,30,31]. Nowadays, the academic community generally supports the results of radioactive geochronological dating method, considering the Yixian Formation as the Early Cretaceous, Barremian–Aptian age [32,33]. The position of biota of the Yixian Formation within the Jehol Biota, which it belongs to, is discussed by Rasnitsyn (2020) [34]. Up to date, there are 5 species of Macroxyelinae reported from Daohugou, and 14 species from Huangbanjigou [35].

The angle between Rs+M and 1-Rs is usually obtuse in Macroxyelinae, but the angle varies among the tribes. The formation of angle between Rs+M and 1-Rs is discussed (red in Figure 3) using a series of Macroxyelinae fossils. Our observations, including the angle of all fossil species of the subfamily Macroxyelinae, are summarized in Table 1. The angle varies from 108° to 145° in the tribe Xyeleciini, from 113° to 154° in the tribe Angaridyelini, 120° in the tribe of Ceroxyelini (only *Ceroxyela* Rasnitsyn, 1966), from 150° to 170° in the tribe of Macroxyelini, and from 145° to 180° in the tribe Gigantoxyelini.

We try to compare and investigate the angle changes between Rs+M and 1-Rs for the tribes in the subfamily Macroxyelinae. We found that there are two trends in the degree of the angle allowing the creation of two informal groups of Macroxyeline tribes (Angaridyelini + Ceroxyelini + Xyeleciini, angle = 108–155°) and (Gigantoxyelini + Macroxyelini, angle = 145–180°). Note that some overlappings of the morphospaces of some tribes are recorded (Figure 4). Interestingly, we found that the first group is also characterized by a short 1-Rs, unlike the second one with generally longer 1-Rs, again with some overlapping. However, we cannot see any correlation between the angle and the fossil age, which is also an interesting observation. Additionally, these fossils show a transition from an obtuse angle toward a perfectly linear alignment of Rs+M and 1-Rs. The angles between Rs+M and 1-Rs became broadened from the tribes Xyeleciini, Angaridyelini and Ceroxyelini to the tribe Macroxyelini, and then to the tribe Gigantoxyelini, while nearly reaching a linear alignment in some species of Macroxyelinae.

The tribe Ceroxyelini was established by Rasnitsyn (1969) initially consisting of only one genus of *Ceroxyela*. Subsequently, Zhang and Zhang (2000) reported two genera, *Sinoxyela* and *Isoxyela*, which were also placed in tribe Ceroxyelini based on their pterostigma of the forewing being sclerotized and basally membranous. In this report, we believe that *Isoxyela* and *Sinoxyela* cannot be attributed to Ceroxyelini primarily based on these different characters: Sc_2_ meeting R before Rs for *Isoxyela* and *Sinoxyela*; the pterostigmal desclerotisation is less extensive than in *Ceroxyela*; the angle between Rs+M and 1-Rs is 120° in *Ceroxyela*, strongly different from those of *Sinoxyela* (155°) and *Isoxyela* (157°); and the length of 1-Rs and the first flagellomere of *Isoxyela* and *Sinoxyela* are obviously different from those of *Ceroxyela*.

The first flagellomere, an important diagnosis trait in Xyelidae [36], is also diversified among the subfamily Macroxyelinae (Figure 5): first flagellomere varies from being three times longer than thread (*Isoxyela* and *Gigantoxyela*), to as long as it (*Scleroxyela*), to 0.6 times shorter than thread (*Leptoxyela*), and up to 0.4 times so (*Ceroxyela*). Based on the above study on the veins and antennae, we believe that the tribe Ceroxyelini may only consist of one genus of *Ceroxyela*, and that the genera *Isoxyela* and *Sinoxyela* should not be included in Ceroxyelini. We would suggest to transfer these two genera to Gigantoxyelini tentatively, based on the sclerotized pterostigma and basal Sc_2_ until more information is available. We also hope that there will be better phylogenetic results in the future to reveal the relationship within Ceroxyelini.

## 5. Conclusions

As the most basal group of Symphyta in Hymenoptera, the family Xyelidae has a special status in the evolutionary history of the Symphyta [7,8,9]. The earliest appearances of xyelids have been reported from the Middle or Upper Triassic of Kyrgyzstan [10], and the Upper Triassic of Australia [11,12], South Africa [13], Argentina [14] and Japan [15]. Xyelidae reached its maximum diversity between the Middle Jurassic and the Early Cretaceous, and its past distribution was much broader than today [6]. Macroxyelinae is a common subfamily of Xyelidae in the Mesozoic [10]. Macroxyelinae encompasses five tribes: Angaridyelini Rasnitsyn, 1967; Ceroxyelini Rasnitsyn, 1969; Gigantoxyelini Rasnitsyn, 1969; Macroxyelini Ashmead, 1898 and Xyeleciini Benson, 1954 [22]. Until now, 28 genera containing 44 fossil species have been reported in Macroxyelinae

We described *Leptoxyela eximia* gen. et sp. nov. and *Scleroxyela cephalota* sp. nov., based on two well-preserved compression fossils from the Lower Cretaceous Yixian Formation and the Middle Jurassic Jiulongshan Formation of China, respectively. *Leptoxyela eximia* gen. et sp. nov. is placed in the tribe Angaridyelini and *S. cephalota* sp. nov. in the tribe Xyeleciini; while both tribes are in Macroxyelinae of Xyelidae. After investigating various angles between Rs+M and 1-Rs for known fossil species of Macroxyelinae, we reported that the angle and the length of 1-Rs are correlated, however, we could not see any correlation between the angles and the fossil ages even within a tribe. In addition, we suggested that *Isoxyela* and *Sinoxyela* should be transferred to Gigantoxyelini and the tribe Ceroxyelini may have only one genus of *Ceroxyela*.

## Figures and Tables

**Figure 1 insects-13-00383-f001:**
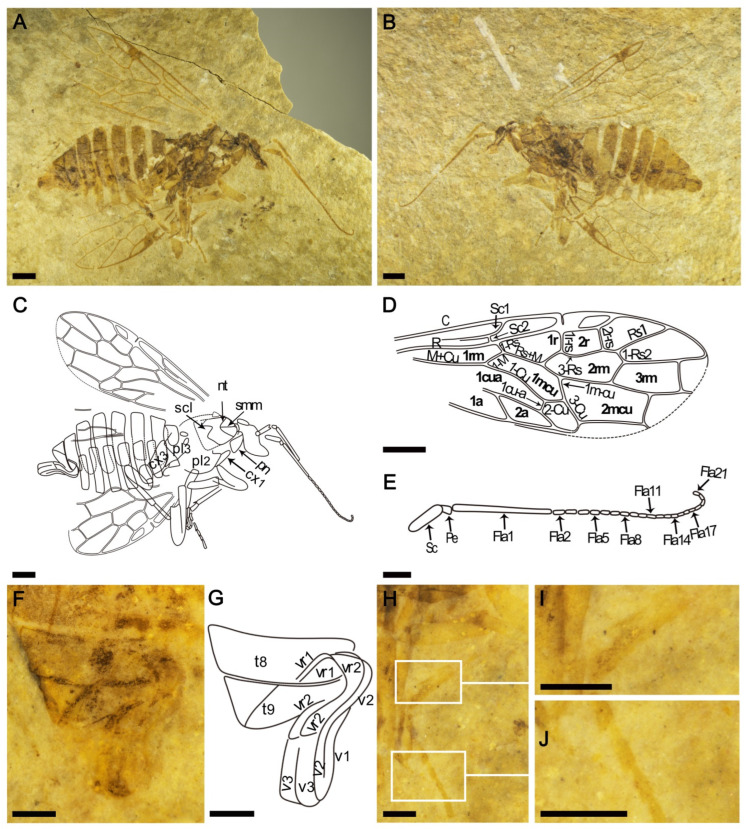
Photographs of *Leptoxyela eximia* gen. et sp. nov., holotype (specimen CNU-HYM-LB-2021002) female. (**A**,**B**), latero-ventral view as preserved. (**C**), line drawing of latero-ventral view. (**D**), line drawing of forewing. (**E**), line drawing of antenna. (**F**), ovipositor under alcohol. (**G**), line drawing of ovipositor. (**H**), middle leg under alcohol. (**I**), middle leg under alcohol, showing leg tibial spurs (white wireframe). (**J**), middle leg under alcohol, showing planter lobes (white wireframe). Scale bars: 1 mm in (**A**–**D**); 0.5 mm in (**E**–**J**). Abbreviations: cx_1_—coxa_1_; cx_3_—coxa_3_; fla_1_–fla_21_—flagellomeres I-XXI; nt—notaulus; pe—pedicel; pl_2_—mesopleuron; pl_3_—metapleuron; pn—pronotum; sc—scape; scl—mesoscutellum; smm—medial mesoscutal suture; t_8_–t_9_—tergum 8–9; v_1_—first valvula; v_2_—second valvula (v_I_ + v_2_—ovipositor); v_3_—third valvula (ovipositor sheath); vr_1_—first valvifer; vr_2_—second valvifer.

**Figure 2 insects-13-00383-f002:**
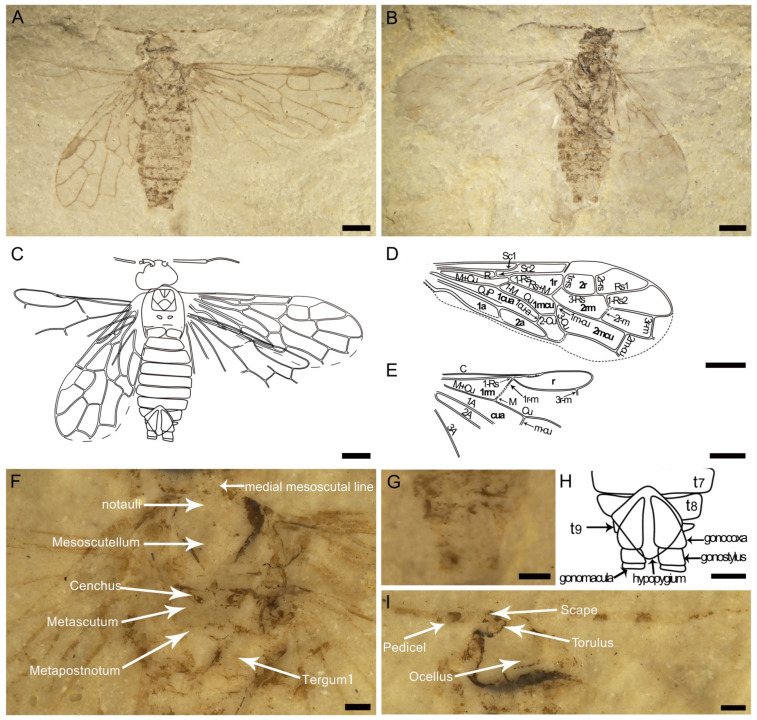
Photographs of *Scleroxyela cephalota* sp. nov., holotype (specimen CNU-HYM-NN-2021004) male. (**A**), dorsal view as preserved. (**B**), ventral view as preserved. (**C**), line drawing of dorsal view with forewings and hind wings. (**D**), line drawing of forewing. (**E**), line drawing of hind wing. (**F**), Photo of body under alcohol. (**G**), Photo of genitalia under alcohol. (**H**), line drawing of genitalia. (**I**). Photo of head under alcohol. Scale bars: 1 mm in (**A**–**E**); 0.25 mm in (**F**–**I**). Abbreviations: t_7_–t_9_, tergum 7–9.

**Figure 3 insects-13-00383-f003:**
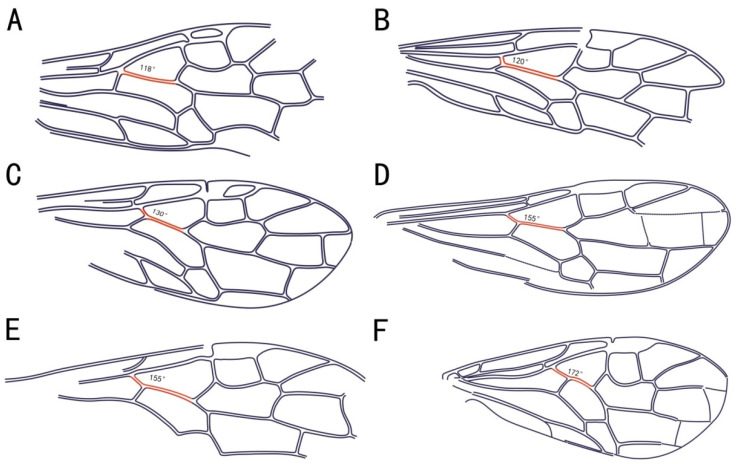
Angle of Rs+M and the section of Rs in the subfamily Macroxyelinae. (**A**), *Uroxyela sicicauda*, Rasnitsyn, 1966 (Xyeleciini). (**B**), *Ceroxyela dolichocera*, Rasnitsyn, 1966 (Ceroxyelini). (**C**), *Leptoxyela eximia*, gen. et sp. nov (Angaridyelini). (**D**), *Megaxyela yaoshanica*, Zhang, 1989 (Macroxyelini). (**E**), *Sinoxyela viriosa*, Zhang and Zhang, 2000 (tribe to be determined). (**F**), *Abrotoxyela multiciliata*, Gao, Ren and Shih, 2009 (Gigantoxyelini).

**Figure 4 insects-13-00383-f004:**
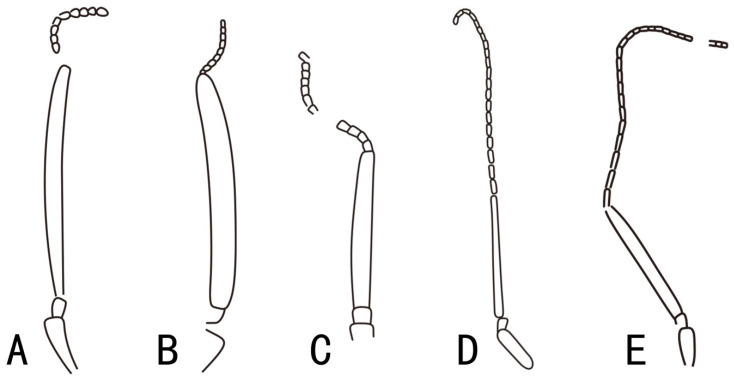
Different types of antennae in the subfamily Macroxyelinae. (**A**), *Isoxyela rudis* Zhang and Zhang, 2000. (**B**), *Gigantoxyela quadrifurcata* Rasnitsyn, 1966. (**C**), *Scleroxyela daohugouensis* Zheng et al., 2021. (**D**), *Leptoxyela eximia* gen. et sp. nov. (**E**), *Ceratoxyela decorosa* Zhang and Zhang, 2000.

**Figure 5 insects-13-00383-f005:**
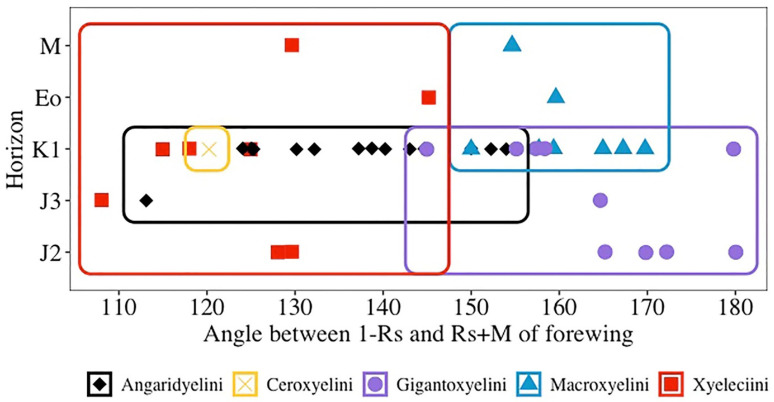
The changes of angle between 1-Rs and Rs+M of forewing for known fossil species of five tribes in Macroxyelinae and the relationship between the angle and fossil age. Angaridyelini in black; Ceroxyelini in yellow; Gigantoxyelini in purple; Macroxyelini in blue; Xyeleciini in red. J_2_—Middle Jurassic; J_3_—Late Jurassic; K_1_—Early Cretaceous; Eo—Eocene; M—Miocene.

**Table 1 insects-13-00383-t001:** Known fossil species of Macroxyelinae.

Tribe	Taxon	Locality and Horizon	Angle between 1-Rs and Rs+M of Forewing	Reference
Xyeleciini	*Bolboxyella bolboica*	E. Siberia, K_1_	-	Rasnitsyn, 1990
	*Microxyelecia brachycera*	Kazakhstan; J_3_	108°	Rasnitsyn, 1969
	*Proxyelia pankowskii*	USA; P	145°	Jouault, Aase and Nel, 2021
	*Scleroxyela cephalota* sp. nov.	China; J_2_	128°	This paper
	*Scleroxyela daohugouensis*	China; J_2_	130°	Zheng et al., 2021
	*Xyelecia xiejiaheensis*	China; N	130°	Hong, 1983
	*Xyelites lingyuanensis*	China; K_1_	125°	Zhang and Zhang, 2000
	*Xyelites trigeminus*	E. Siberia, K_1_	115°	Rasnitsyn, 1966
	*Uroxyela sicicauda*	E. Siberia, K_1_	118°	Rasnitsyn, 1966
Angaridyelini	*Angaridyela chengdeensis*	China; K_1_	143°	Ren, 1995
	*Angaridyela endemica*	China; K_1_	154°	Zhang and Zhang, 2000
	*Angaridyela exculpta*	China; K_1_	139°	Zhang and Zhang, 2000
	*Angaridyela minor*	E. Siberia, K_1_	124°	Rasnitsyn, 1966
	*Angaridyela pallipes*	E. Siberia, K_1_	152°	Rasnitsyn, 1966
	*Angridyela robusta*	China; K_1_	132°	Zhang and Zhang, 2000
	*Angridyela suspecta*	China; K_1_	125°	Zhang and Zhang, 2000
	*Angaridyela vitimica*	E. Siberia, K_1_	125°	Rasnitsyn, 1966
	*Baissoxyela tarsalis*	E. Siberia, K_1_	×	Rasnitsyn, 1969
	*Ceratoxyela decorosa*	China; K_1_	145°	Zhang and Zhang, 2000
	*Leptoxyela eximia* gen. et sp. nov.	China; K_1_	130°	This paper
	*Lethoxyela excurva*	China; K_1_	140°	Zhang and Zhang, 2000
	*Lethoxyela vulgata*	China; K_1_	137°	Zhang and Zhang, 2000
	*Liaoxyela antique*	China; K_1_	150°	Zhang and Zhang, 2000
	*Nigrimonticola longicornis*	Kazakhstan; J_3_	113°	Rasnitsyn, 1966
	*Ophthalmoxyela brachyua*	Kazakhstan; J_3_	×	Rasnitsyn, 1966
Ceroxyelini	*Ceroxyela dolichocera*	E. Siberia, K_1_	120°	Rasnitsyn, 1966
To be determined	*Sinoxyela viriosa*	China; K_1_	155°	Zhang and Zhang, 2000
	*Isoxyela rudis*	China; K_1_	157°	Zhang and Zhang, 2000
Macroxyelini	*Anthoxyela anthophaga*	Russia; K_1_	150°	Rasnitsyn, 1982
	*Anthoxyela baissensis*	Russia; K_1_	170°	Rasnitsyn, 1977
	*Anthoxyela orientalis*	China; K_1_	167°	Gao and Ren, 2008
	*Anthoxyela turgensis*	Russia; K_1_	165°	Rasnitsyn, 1990
	*Brachyoxyela brevinodia*	China; K_1_	159°	Gao, Zhao and Ren, 2011
	*Brachyoxyela gracilenta*	China; K_1_	158°	Gao, Zhao and Ren, 2011
	*Chionoxyela nivea*	E. Siberia, K_2_	-	Rasnitsyn, 1993
	*Megaxyela petrefacta*	USA; N	155°	Brues, 1908
	*Megaxyela yaoshanica*	China; N	155°	Zhang, 1989
	*Paleoxyela nearctica*	USA; P	160°	Jouault, Aase and Nel, 2021
Gigantoxyelini	*Abrotoxyela curva*	China; J_2_	180°	Zheng et al., 2020
	*Abrotoxyela lepida*	China; J_2_	165°	Gao, Ren and Shih, 2009
	*Abrotoxyela multiciliata*	China; J_2_	172°	Gao, Ren and Shih, 2009
	*Chaetoxyela hirsuta*	E. Siberia, K_1_	145°	Rasnitsyn, 1966
	*Gigantoxyela quadrifurcata*	E. Siberia, K_1_	158°	Rasnitsyn, 1966
	*Heteroxyela ignota*	China; K_1_	180°	Zhang and Zhang, 2000
	*Magnaxyela rara*	China; J_2_	170°	Zheng et al., 2020
	*Shartexyela mongolica*	Mongolia; J_3_	165°	Rasnitsyn, 2008

Note. The systematic position of fossil Symphyta in the table was based on the description in original papers, and it does not reflect certain different opinions. J—Jurassic, K—Cretaceous, P—Paleogene, N—Neogene, J_2_—Middle Jurassic, J_3_—Late Jurassic, K_1_—Early Cretaceous.

## Data Availability

All data from this study are available in this paper and the associated papers.

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
