# Peer review of "New Fossil Xyelidae (Hymenoptera: Symphyta) from the Mesozoic of Northeastern China"

_insects, 2022, doi:10.3390/insects13040383_

Round 1

Reviewer 1 Report

Dai et al propose an interesting paper with the description of two new taxa and the investigation of wing venation evolution throughout time and Macroxyelinae tribes. This paper is suitable for publication after consideration of the minor corrections and comments addressed in the attached pdf. I also suggest the authors add a short section in the material and methods explaining the data acquisition for figure 4 and which software was(ere) used. 
Corentin Jouault (MNHN)
02/04/2022

Reviewer 2 Report

This paper describes new fossil taxa from Mesozoic localities in China. It is fairly straightforward; the illustrations are mostly good, but the language should be improved. More specific comments are found in the annotated version of the MS provided.
